# Hearing loss and physical function in the general population: A cross-sectional study

Kaori Daimaru[1], Yukiko Wagatsuma[2]*

1 Graduate School of Comprehensive Human Sciences, University of Tsukuba, Tsukuba, Ibaraki, Japan,
2 Department of Clinical Trial and Clinical Epidemiology, Faculty of Medicine, University of Tsukuba, Tsukuba, Ibaraki, Japan

* ywagats@md.tsukuba.ac.jp

## Abstract

### Objective

Hearing loss is a major public health concern. Higher physical function may be related to the maintenance of hearing acuity. Therefore, this study examined the association between hearing loss and physical function in the general population.

### Methods

This cross-sectional study was conducted with health checkup participants who underwent pure-tone audiometry at a regional health care center in Japan. Information for physical function included handgrip strength, vital capacity (VC), and forced expiratory volume in one second ($FEV_1$). A hearing threshold of >30 dB at 1 kHz and/or >40 dB at 4 kHz in either ear was identified as hearing loss. The characteristics of the subjects were examined with stratification by sex and age group. Multivariable logistic regression analysis was performed to examine the association between hearing loss and physical function with adjustments for age, body mass index and current smoking.

### Results

Among the 4766 study subjects, 56.5% were male. The mean age was 47.7 years (SD: 13.8 years; range: 20–86 years), and the prevalence of hearing loss was 12.8% based on the definition stated above. For females, handgrip strength, VC, and $FEV_1$ showed significant negative associations with hearing loss (multivariable-adjusted OR [95% CI] = 0.691 [0.560–0.852], 0.542 [0.307–0.959], and 0.370 [0.183–0.747], respectively). These associations were not found in males.

### Conclusions

Higher physical function was associated with a lower prevalence of hearing loss among females. This study suggests that it is important to maintain physical function for hearing loss in females. Further studies are required to investigate sex differences in the relationship between physical function and hearing loss in the general population.

**Data Availability Statement:** All relevant data are within the manuscript and its Supporting Information files.

**Funding:** YW received the grant from the Japan Science and Technology Agency (Grant Number

JPMJCE1301). https://www.jst.go.jp/ The funder had no role in study design, data collection and analysis, decision to publish, or preparation of the manuscript.

**Competing interests:** The authors have declared that no competing interests exist.

## Introduction

Hearing function is essential for communicating with others. People with hearing loss encounter difficulties in communication, education, and employment and may experience social isolation [1]. Globally, there are more than 1.5 billion people with some degree of hearing loss, and this number is estimated to increase to 2.5 billion by 2050 [2]. Hearing loss is caused by genetic factors, chronic diseases, noise, and aging [2]. The prevalence of hearing loss increases with increasing age. Middle-aged individuals have a high rate of hypertension and diabetes. These comorbidities have been associated with hearing loss [3]. However, most people are unaware of an early decrease in hearing acuity [4]. Hence, the early detection of hearing difficulties is important to prevent further hearing loss.

Although the mechanism of age-related hearing loss is unclear, oxidative damage from arteriosclerosis has been reported to cause degenerative changes to the stria vascularis of the cochlea of the inner ear [5, 6]. The stria vascularis is rich in capillaries and is a power source for electrical phenomena in the cochlea [5]. Therefore, blood supply to the cochlea is a key to maintaining hearing acuity.

Arteriosclerosis has been reported an association with low physical function [7, 8]. Skeletal muscles, which play a role not only in the locomotive system but also in protein storage and glucose consumption [8], require nutrients and are associated with vascular function [9]. Endothelial dysfunction leads to microvascular dysfunction and decreased microcirculation [10], which could lead to oxidative stress, the inhibition of blood flow, and degenerative changes in the structures of the cochlea [5]. Owing to this mechanism, low skeletal muscle strength reflected to declined physical function may enhance the progression of age-related hearing loss.

In addition, supplying oxygen to vessels is also important for the maintenance of blood flow in the cochlea. A previous study showed that a significant factor in the development of arteriosclerosis is hypoxia in the arterial wall [11]. Forced expiratory volume in one second ($FEV_1$), a respiratory function indicator, has been reported to be positively associated with peak oxygen uptake ($VO_{2peak}$) [12]. Participants who had low $VO_{2peak}$ showed a greater hearing threshold than those with high $VO_{2peak}$ [13]. Another study showed that $FEV_1$ had an inverse association with the risk of coronary artery disease caused by arteriosclerosis [14]. From these findings, low respiratory function may damage vascular conditions in the cochlea and affect age-related hearing loss.

Most previous studies on hearing loss and physical function have examined whether hearing loss affects physical function and performance in elderly individuals [15–17]. In an opposite direction, few studies have been conducted to examine the effect of physical function on hearing loss. Since the status of physical function develops gradually and is influenced by lifestyle habits, it is important to examine the relationship between hearing loss and related risks among individuals, including young and middle-aged individuals, in the general population. Physical function capacity in early adulthood may affect vascular conditions and hearing acuity later in life. Therefore, this study aimed to examine the association between hearing loss (dependent variable) and physical function in the general population, among participants in health checkups.

## Materials and methods

### Study area and subjects

This study was conducted among health checkup participants in a regional health care center located in Mito City and its outreach sites in Japan. Community dwellers and workers in Japan

are recommended to undergo health checkups once a year to detect diseases early. Individuals older than 40 years are recommended to undergo health checkups to prevent lifestyle-related diseases.

We asked all health checkup participants to join this study. We enrolled all participants who provided written informed consent, were 20 years old and above, and underwent pure-tone audiometry at the health checkups. Health checkup information was collected between April 2018 and March 2020. Information including demographic characteristics, anthropometric characteristics, blood chemistry results, audiometry results, physical function, medications, comorbidities, medical history, and lifestyle habits was obtained. For the participants who underwent pure-tone audiometry in both 2018 and 2019, information from 2018 was used.

### Measurements

**Audiometry.**   In this study, audiometry was conducted according to the National Health Examination Guideline of Japan (Industrial Safety and Health Act, Ordinance on Industrial Safety and Health, Articles 43, 44, 45) at the regional health care center and outreach locations. Air conduction hearing thresholds for each ear were measured using pure-tone audiometry [18] at 1 and 4 kHz with two types of audiometers (AA-57, AA-58, Rion Co. Ltd., Japan) by trained staff. The measurements were taken in a soundproof booth or by using headphones for a quiet environment. Bone conduction hearing threshold tests were not performed. The frequency ranges of 1 kHz and 4kHz were used for everyday speech and early age-related hearing loss, respectively. One type of audiometer (AA-57) was used for lifestyle-related disease examinations and indicated whether an abnormality was present. Another audiometer (AA-58) was used for detailed medical examinations, which presented hearing thresholds for each ear at 1 and 4 kHz. A hearing threshold of >30 dB at 1 kHz and/or >40 dB at 4 kHz in either ear was identified as hearing loss, in line with the national guidelines. The study subjects were divided into two groups based on the definition of hearing loss: the hearing loss group and the no hearing loss group.

**Physical function.**   Handgrip strength and respiratory function were used as physical function indices. Handgrip strength is a marker of muscle strength [19], and respiratory function is related to respiratory muscle strength and is associated with frailty, features of slow gait speed, and low physical activity [20].

Handgrip strength was assessed using a Smedley digital grip dynamometer (T.K.K.5401 GRIP D, Takei Scientific Instruments, Japan). Study subjects were instructed to be in an upright position with their arms naturally lowered and to grasp the Smedley digital grip dynamometer as hard as possible without touching their body or clothing. Two measurements were taken alternating the right and left sides, and the average of the maximum values for each side displayed on the grip dynamometer was used. Vital capacity (VC) and forced expiratory volume in one second ($FEV_1$) were used as respiratory function indices. Both parameters were measured using a spirometer (FDAC-7D, Fukuda Denshi, Japan). Trained staff instructed the study subjects to inhale and then exhale.

**Other measurements.**   Height and weight were measured using an automatic height meter (DC250, Tanita, Japan). Body mass index (BMI) was calculated as weight divided by height squared. Diastolic and systolic blood pressures were measured using an automatic sphygmomanometer (MPV-3301, Nihon Kohden Corporation, Japan) on the arm in the resting position. Hypertension was defined as a systolic blood pressure $\geq$ 140 mmHg, a diastolic blood pressure $\geq$ 90 mmHg, or the self-reported intake of medication for hypertension in a questionnaire [21]. Fasting blood glucose, HbA1c, low-density lipoprotein (LDL) cholesterol,

high-density lipoprotein (HDL) cholesterol, and triglyceride levels were measured at a regional health care center. Diabetes was defined as a fasting glucose level $\geq$ 126 mg/dL and an HbA1c level $\geq$ 6.5% or as the self-reported intake of medication for diabetes in a questionnaire [22]. Dyslipidemia was defined as an LDL cholesterol level $\geq$ 140 mg/dL, an HDL cholesterol level < 40 mg/dL, a triglyceride level $\geq$ 150 mg/dL, or the self-reported intake of medication for dyslipidemia [23].

**Questionnaire.** Information on current medications for hypertension, diabetes, and dyslipidemia, medical histories of heart disease and stroke, and lifestyle habits including current smoking, alcohol consumption, and regular exercise was obtained using a self-reported standardized questionnaire for lifestyle-related diseases specified by the Ministry of Health, Labour and Welfare of Japan. Alcohol consumption was defined as drinking every day, and regular exercise was defined as exercise for > 30 min/day and > 2 times/week.

## Statistical analysis

The characteristics of the study subjects were summarized in terms of age, sex, health checkup course, BMI, percent predicted VC (%VC), $FEV_1$/forced VC (FVC), the prevalence of hearing loss, and the rates of comorbidities and lifestyle habits. Continuous variables are presented as the mean values and standard deviations (SDs), and categorical variables are presented as frequencies and percentages. The differences between the two groups were tested using the t test for continuous variables and the chi-square test or Kendall's tau-b test for categorical variables. After stratifying by sex and age group (<40, 40–49, 50–59, $\geq$60 years), the characteristics of the study subjects were further summarized for hearing loss, physical function and comorbidities.

Univariate analyses examined the association between hearing loss and each covariate. Male sex was significantly associated with hearing loss when females were used as a reference. Moreover, lifestyle habits such as current smoking and alcohol consumption were significantly associated with hearing loss in males but showed no significant association in females (S1 Table). Due to such differences by sex, further analyses were conducted with stratification by sex.

Multivariable logistic regression models were used to calculate the adjusted odds ratio (aOR) and 95% confidence interval (CI) for the relationship between hearing loss and each physical function, stratified by sex. Age was used as a continuous variable. The interaction terms between physical function and the covariates included in each model were examined. The correlation matrix of multivariable parameter estimates was examined to determine if there were serious problems with multicollinearity. BMI was entered into the multivariable analysis because BMI affects physical function as well as hearing loss, and the association between physical function and BMI varies by sex (i.e., some men with high BMI have high muscle mass) [24–26]. Current smoking was added because it has been reported to be a risk factor for hearing loss [27] and physical function [28]. For sensitivity analysis, other covariates, such as hypertension, diabetes and alcohol consumption, were examined. Age squared was also considered in the model to examine the nonlinear relationship of hearing loss with age.

IBM SPSS Statistics version 27 (IBM, USA) was used for all analyses. Statistical significance was defined as a two-sided $p$ value < 0.05.

**Sample size.** The prevalence of hearing loss in the high and low groups divided by the median handgrip strength was assumed to be 15.4% and 20.2%, respectively, based on a preliminary analysis using 2018 data. The required sample size was calculated to be 1888 in the two groups combined with a 5% significance level, an 80% power, and a two-tailed test. When stratified analysis by sex was considered, the required sample size doubled to 3776. Assuming a missing rate of 20%, the required sample size was 4720.

**Ethics.** This study was approved by the Ethics Review Committee of the Faculty of Medicine at the University of Tsukuba, Japan. Written informed consent was obtained from all participants. All procedures were performed in accordance with the Declaration of Helsinki.

## Results

The numbers of participants who underwent health checkup examinations in 2018 and 2019 were 4270 and 4014, respectively. In 2019, the number of new participants was 1503. The total number of the participants in 2018 and the new participants in 2019 was 5773. Among these participants, those who had not undergone audiometry from April 2018 to March 2020 (n = 977) and those who were under 20 years of age (n = 30) were excluded. The total number of study subjects was 4766 (coverage rate was 82.6% (4766/5773)). Handgrip strength was measured for 4572 subjects, and spirometry for respiratory function was conducted for 1683 subjects (S1 Data).

Table 1 shows the characteristics of the study subjects. The study sample was 56.5% male, and the mean age was 47.7 years (SD: 13.8 years; range: 20–86 years). The overall rate of detailed medical examinations, including numerical values of audiometry, was 35.0% (1670/4766). The overall %VC $< 80$% and $FEV_1$/FVC $< 70$% were 3.6% (60/1683) and 8.3% (140/1682), respectively. The overall prevalence of hearing loss was 12.8% (609/4766). The prevalence of hearing loss, an $FEV_1$/FVC $< 70$%, and the rates of comorbidities for hypertension, diabetes and dyslipidemia were higher in males than in females (all with $p < 0.001$). The overall rates of current smoking, alcohol consumption, and regular exercise were 23.8% (1068/4482), 21.9% (977/4465), and 23.0% (1028/4479), respectively. These values were higher in males than in females (all with $p < 0.001$).

Table 2 shows hearing loss, physical function and comorbidities by age group stratified by sex. Handgrip strength showed a peak value in the 40-49-year age group, then declined as age increased for both sexes ($p<0.001$). The decline was slightly smaller in females than males (-3.3 kg in males vs. -2.3 kg in females). For respiratory function measured by VC and FEV1, the peak value was observed in the youngest age group (aged $<40$ years), and the values declined as age increased in both sexes ($p<0.001$). The histogram of each physical function by sex indicated a normal distribution (S1 Fig). The rates for hypertension and diabetes increased as age increased and among the subjects aged $\geq60$ years, 63.2% of the males and 53.2% of the females had hypertension. The prevalence of diabetes was much lower than that of hypertension. The highest rate in the group aged $\geq60$ years was 13.8% for males and 7.9% for females.

When physical function groups were divided using sex-specific median values, the high physical function group showed a lower prevalence of hearing loss, lower rates of comorbidities for hypertension and diabetes, and a lower rate of a history of heart disease than the respective low physical function groups ($p <0.05$; S2 Table).

Table 3 shows the results of the multivariable logistic regression analysis stratified by sex. Handgrip strength per 5 kg (handgrip strength/5 kg) showed a significant negative association with hearing loss after adjusting for age (continuous variable), BMI and current smoking in females (aOR [95% CI] = 0.691 [0.560–0.852]). In males, handgrip strength did not show a significant relationship with hearing loss. Similarly, for respiratory function, VC and $FEV_1$ showed a significant association with hearing loss only in females (aOR [95% CI] = 0.542 [0.307–0.959], 0.370 [0.183–0.747], respectively; Table 3). For sensitivity analyses, other covariates were considered and included in the models. After adding comorbidities of hypertension and diabetes in the models, the association of handgrip strength with hearing loss remained significant in females (aOR [95% CI] = 0.691 [0.560–0.863]). We further tested the modification effect of comorbidities; however, no interaction with physical function was observed. We

**Table 1. Characteristics of study subjects (n = 4766).**

| Variable | Overall (n = 4766) | | Male (n = 2674) | | Female (n = 2072) | | p value |
|---|---|---|---|---|---|---|---|
| | n | | n | | n | | |
| Age, years, mean, SD | 4766 | 47.7, 13.8 | 2694 | 47.5, 13.8 | 2072 | 48.0, 13.9 | 0.260 |
| Age group, years, n (%) | 4766 | | 2694 | | 2072 | | |
| 20–29 | | 576 (12.1) | | 310 (11.5) | | 266 (12.8) | 0.136 |
| 30–39 | | 839 (17.6) | | 516 (19.2) | | 323 (15.6) | 0.831 |
| 40–49 | | 1056 (22.2) | | 610 (22.6) | | 446 (21.5) | <0.001 |
| 50–59 | | 1236 (25.9) | | 635 (23.6) | | 601 (29.0) | 0.180 |
| 60–69 | | 836 (17.5) | | 503 (18.7) | | 333 (16.1) | 0.787 |
| 70–86 | | 223 (4.7) | | 120 (4.5) | | 103 (5.0) | 0.728 |
| Health checkup course, n (%) | 4766 | | 2694 | | 2072 | | |
| Detailed medical examination | | 1670 (35.0) | | 1007 (37.4) | | 663 (32.0) | <0.001 |
| Lifestyle-related diseases examination | | 2621 (55.0) | | 1456 (54.0) | | 1165 (56.2) | 0.134 |
| Mandatory medical examination | | 348 (7.3) | | 174 (6.5) | | 174 (8.4) | 0.011 |
| Others | | 127 (2.7) | | 57 (2.1) | | 70 (3.4) | 0.007 |
| BMI, kg/m$^2$, mean, SD | 4766 | 23.7, 4.0 | 2694 | 24.6, 3.9 | 2072 | 22.6, 4.0 | <0.001 |
| %VC, < 80%, n (%) | 1683 | 60 (3.6) | 1014 | 40 (3.9) | 669 | 20 (3.0) | 0.301 |
| FEV$_1$/FVC, < 70%, n (%) | 1682 | 140 (8.3) | 1013 | 109 (10.8) | 669 | 31 (4.6) | <0.001 |
| Hearing loss [a], n (%) | 4766 | 609 (12.8) | 2694 | 413 (15.3) | 2072 | 196 (9.5) | <0.001 |
| Comorbidities | | | | | | | |
| Hypertension, n (%) | 4766 | 1734 (36.4) | 2694 | 1153 (42.8) | 2072 | 581 (28.0) | <0.001 |
| Diabetes, n (%) | 4492 | 286 (6.4) | 2576 | 217 (8.4) | 1916 | 69 (3.6) | <0.001 |
| Dyslipidemia, n (%) | 4673 | 2176 (46.6) | 2645 | 1382 (52.2) | 2028 | 794 (39.2) | <0.001 |
| Lifestyles | | | | | | | |
| Current smoking, n (%) | 4482 | 1068 (23.8) | 2570 | 860 (33.5) | 1912 | 208 (10.9) | <0.001 |
| Alcohol consumption [b], n (%) | 4465 | 977 (21.9) | 2561 | 804 (31.4) | 1904 | 173 (9.1) | <0.001 |
| Regular exercise [c], n (%) | 4479 | 1028 (23.0) | 2568 | 701 (27.3) | 1911 | 327 (17.1) | <0.001 |

Abbreviations: SD, standard deviation; BMI, body mass index; %VC, percentage of the predicted vital capacity; FEV$_1$, forced expiratory volume in one second; FVC, forced vital capacity.

a) Defined as hearing threshold of > 30 dB at 1 kHz and/or > 40 dB at 4 kHz in either ear with pure-tone audiometry.

b) Defined as drinking every day, c) >30 min/day, > 2 times/week, P value was calculated using t test for continuous variable and chi-square test for categorical variable.

also included height to adjust for respiratory function in the models, but it was not significant for VC and FEV1. Age squared was also considered to examine the nonlinear relationship of hearing loss with age. Only a model for handgrip strength in females showed a significant effect of this variable (p = 0.023), and the significance of the associations between each physical function and hearing loss remained the same.

## Discussion

The early identification of hearing loss and the modification of the status of chronic diseases and lifestyle patterns may improve hearing acuity later in life [29]. The current study used the health checkup information of individuals over 20 years old. This study found that higher physical function, which was represented by handgrip strength and respiratory function, was associated with a lower prevalence of hearing loss in females. This result was shown after controlling for age, BMI, and current smoking. This result indicates sex discrepancies in the association between physical function and hearing loss.

**Table 2. Hearing loss, physical function and comorbidities stratified by age and sex (n = 4766).**

| Variable | Overall (n = 2694) | <40 years (n = 826) | | 40–49 years (n = 610) | | 50–59 years (n = 635) | | ≥ 60 years (n = 623) | | p value [a] |
|---|---|---|---|---|---|---|---|---|---|---|
| | n | n | | n | | n | | n | | |
| **Males (n = 2694)** | | | | | | | | | | |
| Hearing loss [b], n (%) | 2694 | 826 | 21 (2.5) | 610 | 37 (6.1) | 635 | 99(15.6) | 623 | 256(41.1) | <0.001 |
| Physical function | | | | | | | | | | |
| Handgrip strength, kg, mean (SD) | 2601 | 803 | 40.7(6.6) | 593 | 41.3(6.4) | 613 | 40.7(5.9) | 592 | 38.0(5.7) | <0.001 |
| VC, L, mean (SD) | 1014 | 118 | 4.77(0.58) | 220 | 4.59(0.62) | 260 | 4.23(0.63) | 416 | 3.80(0.57) | <0.001 |
| $FEV_1$, L, mean (SD) | 1014 | 118 | 4.00(0.55) | 220 | 3.67(0.51) | 260 | 3.24(0.50) | 416 | 2.82(0.50) | <0.001 |
| Comorbidities | | | | | | | | | | |
| Hypertension, n (%) | 2694 | 826 | 159 (19.2) | 610 | 231 (37.9) | 635 | 369(58.1) | 623 | 394(63.2) | <0.001 |
| Diabetes, n (%) | 2576 | 724 | 13 (1.8) | 604 | 47 (7.8) | 631 | 72(11.4) | 617 | 85(13.8) | <0.001 |
| Variable | Overall (n = 2072) | <40 years (n = 589) | | 40–49 years (n = 446) | | 50–59 years (n = 601) | | ≥ 60 years (n = 436) | | p value [a] |
| | n | n | | n | | n | | n | | |
| **Females (n = 2072)** | | | | | | | | | | |
| Hearing loss [b], n (%) | 2072 | 589 | 16 (2.7) | 446 | 16 (3.6) | 601 | 42(7.0) | 436 | 122(28.0) | <0.001 |
| Physical function | | | | | | | | | | |
| Handgrip strength, kg, mean (SD) | 1971 | 575 | 24.6(4.4) | 421 | 25.5(4.2) | 569 | 24.4(3.7) | 406 | 23.2(4.1) | <0.001 |
| VC, L, mean (SD) | 669 | 55 | 3.29(0.51) | 143 | 3.18(0.45) | 196 | 2.99(0.42) | 275 | 2.62(0.41) | <0.001 |
| $FEV_1$, L, mean (SD) | 669 | 55 | 2.82(0.43) | 143 | 2.61(0.40) | 196 | 2.36(0.36) | 275 | 2.01(0.33) | <0.001 |
| Comorbidities | | | | | | | | | | |
| Hypertension, n (%) | 2072 | 589 | 40 (6.8) | 446 | 91 (20.4) | 601 | 218(36.3) | 436 | 232(53.2) | <0.001 |
| Diabetes, n (%) | 1916 | 443 | 4 (0.9) | 444 | 11 (2.5) | 597 | 20(3.4) | 432 | 34(7.9) | <0.001 |

Abbreviations: SD, standard deviation; VC, vital capacity; $FEV_1$, forced expiratory volume in one second.

a) P value was calculated by Kendall's tau-b test for categorical variables and by ANOVA for continuous variables.

b) Defined as hearing threshold of >30 dB at 1 kHz and/or > 40 dB at 4 kHz in either ear with pure-tone audiometry.

One explanation would be that the association between physical function and hearing loss is different due to biological mechanisms by sex. A review summarized the clinical features of sex differences in acquired sensorineural hearing loss, comparing animal investigations of cochlear sexual dimorphism [30]. Sex differences in hearing loss have been discussed in relation to the protective effect of estrogen [31]. Increasing evidence has suggested that the proinflammatory response in otolaryngological diseases is linked to the level of sex hormones [32]. The vessels of female rats constrict less and relax more in response to adrenergic stimulation than do the vessels of males [33]. In the current study, half of the women were over 50 years old. Menopausal transition occurs around the ages of 46–52 years, and this was the peak frequency age of the current study. In women, muscle strength starts declining during early postmenopausal ages, earlier than in men [34]. Moreover, testosterone has been reported to have a negative effect on hearing [35]. The diminishing levels of testosterone in the elderly males may help their hearing, compared to the aged females, while muscle strength declines by aging. It might be difficult to detect the association with hearing loss by the declining pattern of physical function, accompanied by hormonal decline.

Handgrip strength/5kg, VC/L, and FEV1/L in females were associated with 31%, 46%, and 63% reduced risks of hearing loss, though the results in males were not significant in this study. The effect size of females in this study is thought to be comparable with those of previous studies showing significant association between hearing loss and physical function in

**Table 3. Adjusted odds ratios for hearing loss with physical functions stratified by sex.**

a) Handgrip strength

Males (n = 2483)

| Variable | n | aOR [95% CI] | p value |
|---|---|---|---|
| Handgrip strength/5kg | 2483 | 1.030 [0.927–1.145] | 0.579 |
| Age, years | 2483 | 1.121 [1.106–1.136] | **<0.001** |
| Body mass index, kg/m$^2$ | 2483 | 1.028 [0.992–1.066] | 0.124 |
| Current smoking, yes/no | 827/2483 | 1.222 [0.935–1.597] | 0.142 |

Prevalence of hearing loss: 15.5% (386/2483)

Females (n = 1820)

| Variable | n | aOR [95% CI] | p value |
|---|---|---|---|
| Handgrip strength/5kg | 1820 | 0.691 [0.560–0.852] | **0.001** |
| Age, years | 1820 | 1.092 [1.074–1.110] | **<0.001** |
| Body mass index, kg/m$^2$ | 1820 | 1.012 [0.969–1.056] | 0.603 |
| Current smoking, yes/no | 202/1820 | 1.167 [0.681–2.001] | 0.575 |

Prevalence of hearing loss: 10.1% (183/1820)

b) VC

Males (n = 1014)

| Variable | n | aOR [95% CI] | p value |
|---|---|---|---|
| VC, L | 1014 | 0.897 [0.665–1.210] | 0.477 |
| Age, years | 1014 | 1.132 [1.106–1.158] | **<0.001** |
| Body mass index, kg/m$^2$ | 1014 | 1.091 [1.036–1.149] | **0.001** |
| Current smoking, yes/no | 239/1014 | 1.577 [1.045–2.380] | **0.030** |

Prevalence of hearing loss: 24.4% (247/1014)

Females (n = 669)

| Variable | n | aOR [95% CI] | p value |
|---|---|---|---|
| VC, L | 669 | 0.542 [0.307–0.959] | **0.035** |
| Age, years | 669 | 1.083 [1.055–1.112] | **<0.001** |
| Body mass index, kg/m$^2$ | 669 | 1.036 [0.978–1.097] | 0.230 |
| Current smoking, yes/no | 39/669 | 1.046 [0.380–2.877] | 0.930 |

Prevalence of hearing loss: 17.0% (114/669)

c) FEV$_1$

Males (n = 1014)

| Variable | n | aOR [95% CI] | p value |
|---|---|---|---|
| FEV$_1$, L | 1014 | 0.929 [0.650–1.326] | 0.684 |
| Age, years | 1014 | 1.133 [1.106–1.161] | **<0.001** |
| Body mass index, kg/m$^2$ | 1014 | 1.092 [1.038–1.150] | **0.001** |
| Current smoking, yes/no | 239/1014 | 1.569 [1.035–2.377] | **0.034** |

Prevalence of hearing loss: 24.4% (247/1014)

Females (n = 669)

| Variable | n | aOR [95% CI] | p value |
|---|---|---|---|
| FEV$_1$, L | 669 | 0.370 [0.183–0.747] | **0.006** |
| Age, years | 669 | 1.074 [1.044–1.104] | **<0.001** |
| Body mass index, kg/m$^2$ | 669 | 1.035[0.977–1.097] | 0.242 |
| Current smoking, yes/no | 39/669 | 0.965 [0.351–2.659] | 0.946 |

Prevalence of hearing loss: 17.0% (114/669)

Abbreviations: aOR, adjusted odds ratio; 95% CI, 95% confidence interval; VC, vital capacity; FEV$_1$, forced expiratory volume in one second.

Hearing loss was defined as hearing threshold of > 30 dB at 1 kHz and/or > 40 dB at 4 kHz in either ear with pure-tone audiometry. Adjusted odds ratios with 95% CI and p value were calculated using multivariable logistic regression analysis.

cross-sectional design, despite that the assumed causal direction between hearing loss and physical function in the present study was converse to previous studies, i.e., physical function as a dependent variable [15–17, 36]. However, there are some differences among this study and previous studies. First, the results in this study were analyzed with stratification by sex, while no stratified analysis was reported in previous studies. Second, physical function parameters used in prior studies were different from this study, such as ADL, lower extremity mobility [15], self-reported and accelerometer-measured physical activity [16], and speed and gait composite score [17].

Most previous studies mainly investigated the association in elderly people (e.g., those over 60 years old). Few studies have examined this association in the general population. A cross-sectional study showed that low handgrip strength was associated with self-reported hearing loss in individuals over 60 years old [36]. A prospective study reported a significant effect of physical fitness on hearing loss using composite z scores from handgrip strength tests and four other tests, including vertical jump height, single-leg balance with eye closed, standing forward bending, and whole-body reaction time [37]. They used composite scores from five tests and showed a dose-response effect on hearing loss. However, the results were not consistent for each test, and handgrip strength did not show an effect on hearing loss. Handgrip strength reflects muscle strength, while other physical fitness tests focus more on balance, power, flexibility and reaction time.

This study used two different physical function indicators, namely, handgrip strength and respiratory function. Handgrip strength reflects muscle strength and has been reported to be associated with age-related disorders. A systematic review on sensory impairment and frailty determined with handgrip strength and other factors indicated cross-sectional associations between hearing loss and prefrailty and frailty [38]. Another systematic review showed that chronic obstructive pulmonary disease (COPD) patients with low $FEV_1$ had a higher incidence of hearing loss than controls [39]. Previous studies reported muscle strength and respiratory function separately. The current study presented both types of physical function to examine the association with hearing loss.

The high physical function group showed lower rates of hypertension, diabetes, dyslipidemia, and history of heart disease and stroke than the low physical function group. These findings suggest that individuals with high physical function have low risk factors for hearing loss [40]. This difference in characteristics may contribute to the difference in the prevalence of hearing loss between the high and low physical function groups.

Regarding the biological plausibility of the findings, higher physical function may promote microcirculation and blood flow in the cochlea of the inner ear and may result in the improvement of hearing loss [5, 6]. However, the current study could not clarify the causal relationship between hearing loss and physical function because of its cross-sectional design.

This study defined hearing loss as a threshold of >30 dB at 1 kHz and/or >40 dB at 4 kHz. The prevalence of hearing loss in this study, including 4766 health checkup subjects, was 12.8%. The prevalence of hearing loss was in the range of values reported in previous studies [41, 42]. The WHO indicates normal hearing as less than 20 dB in a better hearing ear [2]. This threshold is lower than that in the current study. A threshold of 20 dB may be more appropriate for detecting early grades of hearing loss. In this study, a health checkup with audiometry data was recorded as a dichotomous variable (yes or no) according to the Japanese health checkup guidelines. Thus, this study could not use exact dB levels in the analysis.

The strength of this study is the inclusion of a general population with a wide age range among community dwellers. Therefore, this general population is suitable for exploring early hearing loss. Our sample size of approximately 5000 was large enough to examine the association by using stratified analysis. However, this study had some limitations. First, the causality

between hearing loss and physical function could not be assessed because of the study's cross-sectional design. Second, information on the cause of hearing loss in each subject was not obtained. Hearing loss caused by factors other than aging could be included. Third, we used self-reported information on medication and lifestyle habits, which could have been overestimated or underestimated. Last, the study used the mean value of the highest values of the forces of each hand for handgrip strength. Since this was a factory setting, the study was not able to record the maximum value of the dominant hand, which would be more relevant to examine the relationship of handgrip strength with hearing loss.

## Conclusion

This study found an association between hearing loss and physical function in a population undergoing regional health checkups. Higher physical function in handgrip strength and respiratory function were independently associated with a lower prevalence of hearing loss in females. This study suggests the importance of maintaining physical function for hearing loss in females. Sex discrepancies in the association between physical function and hearing loss need to be investigated in future studies. It is necessary to study the long-term effect of physical function at a younger age on hearing loss later in life.

## Supporting information

**S1 Table. Crude odds ratios for association between hearing loss and each variable.**
(DOCX)

**S2 Table. Characteristics of study subjects in high and low physical function groups.**
(DOCX)

**S1 Fig. Histograms of physical function by sex.**
(PDF)

**S1 Data.**
(XLSX)

## Acknowledgments

Authors are grateful to the participants in the study, doctors, nurses, and staff of Mito-Kyodo General Hospital.

## Author Contributions

**Conceptualization:** Kaori Daimaru, Yukiko Wagatsuma.

**Data curation:** Kaori Daimaru, Yukiko Wagatsuma.

**Formal analysis:** Kaori Daimaru.

**Funding acquisition:** Yukiko Wagatsuma.

**Investigation:** Kaori Daimaru.

**Methodology:** Kaori Daimaru, Yukiko Wagatsuma.

**Project administration:** Yukiko Wagatsuma.

**Resources:** Yukiko Wagatsuma.

**Supervision:** Yukiko Wagatsuma.

**Validation:** Yukiko Wagatsuma.

**Writing – original draft:** Kaori Daimaru.

**Writing – review & editing:** Kaori Daimaru, Yukiko Wagatsuma.

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
