## [Decision Letter · Decision Letter 0]

31 May 2022

PONE-D-22-09462Hearing loss and physical function in the general populationPLOS ONE

Dear Dr. Wagatsuma,

Thank you for submitting your manuscript to PLOS ONE. After careful consideration, we feel that it has merit but does not fully meet PLOS ONE’s publication criteria as it currently stands. Therefore, we invite you to submit a revised version of the manuscript that addresses the points raised during the review process. Please address all points raised by the two reviewers,

We look forward to receiving your revised manuscript.

Kind regards,

Karin Bammann, Ph.D.

Academic Editor

PLOS ONE

Journal Requirements:

Additional Editor Comments:

Please carefully address all points raised by the reviewers.

Reviewers' comments:

Reviewer's Responses to Questions

**Comments to the Author**

1. Is the manuscript technically sound, and do the data support the conclusions?

Reviewer #1: Partly

Reviewer #2: Yes

2. Has the statistical analysis been performed appropriately and rigorously? 

Reviewer #1: No

Reviewer #2: Yes

3. Have the authors made all data underlying the findings in their manuscript fully available?

Reviewer #1: Yes

Reviewer #2: Yes

4. Is the manuscript presented in an intelligible fashion and written in standard English?

Reviewer #1: Yes

Reviewer #2: Yes

5. Review Comments to the Author

Reviewer #1: The submitted manuscript investigates the association between physical function and hearing loss in a Japanese population aged 20-86 years old. The study has great potential, especially due to the underlying data set. However, there are some major revisions required regarding the analyses.

Title

1. Please indicate the study design in the title.

Abstract

2. The mention of physical activity in the objectives is slightly misleading. The focus should be on physical function.

Introduction

3. Please indicate in ll. 50-51 which population these numbers refer to.

4. A few more references in the first paragraph would be appropriate.

5. The introduction focusses primarily on the association of arteriosclerosis and hearing loss. Simultaneously, research results on physical function come short, especially regarding respiratory function. Please elaborate the results of the referenced studies.

6. It remains unclear, why it is important to investigate this association and why it should be investigated in a younger population.

Methods

7. If I understand correctly, you used the average of the maximum values of grip strength in the left and right hand. Instead, using the overall maximum or the maximum of the dominant hand is more appropriate.

8. I suggest moving the definitions to the respective measurement descriptions.

9. Please describe how covariates were tested for multicollinearity.

10. It is described that hypertension, diabetes, current smoking, and alcohol consumption were added to the models. However, only hypertension is in all three models while diabetes, current smoking, and alcohol consumption were only included for handgrip strength. In addition, variables vary in the age-stratified analyses and this is not further described. Please specify how your models were derived.

Results

11. Please indicate how many of the participants were excluded due to not undergoing audiometry or age restrictions and add the response rate.

12. Please add the unit to the standard deviations.

13. The definition of lifestyle variables is only provided in Table 1 and is missing in the methods section.

14. All Tables: Overall n in the top row is missing.

15. Table 1: Please also state the number of male and female participants in the respective columns.

16. Table 1: Instead of reporting medication intake, it would be more insightful to report how many were defined as having hypertension, diabetes, or dyslipidemia.

17. Supplementary material should be referred to only briefly in the results section.

18. Table 2 does not provide highly relevant information and should be moved to the supplements. Instead, it would be interesting to report physical function and hearing loss by age groups. This could also replace S2.

19. Table 3: Due to risk of residual confounding, age should not be included in the analyses as a binary variable but continuous.

20. Table 3: Participant’s sex appears to be highly relevant. Please consider stratifying your analyses by sex.

21. Table 4: The age group of over 50-year-olds is still highly heterogeneous regarding physical function and hearing loss. In this context, the age stratification does not seem very relevant. As suggested above, stratifying for sex and including age continuously would deliver more meaningful results.

Discussion

22. In ll. 259-361, it reads as if physical activity is equated with physical function. Please put this into context.

23. What were the findings of ref. 29? (ll. 361-363)

24. It remains unclear, why associations between physical function parameters and other health outcomes are reported (ll. 370-374).

25. How do you explain your unsignificant findings in the younger age group?

Reviewer #2: Thanks for the opportunity to comment on this manuscript. This study is quite interesting which investigated the associations between physical function and hearing loss. Some specific brief comments:

Abstract:

• In the method and results of the abstract, the definition of hearing loss should be added.

Methods:

• Details about how the audiometric test was conducted should be provided.

• Is bone-conduction used in audiometric testing or not?

• The definition of hearing loss is >30 dB HL in the methods, whereas in table 1 it says “Defined as hearing threshold of > 30 dB at 1 kHz and/or > 40 dB at 4 kHz in either ear with pure-tone audiometry”. It is a bit confusing.

• Can you please explain why BMI and smoking were not adjusted for when looking at the associations between VC and FEV1 and hearing loss?

Results:

• When looking at the association between FEV1 and hearing loss in ALL participants, the aOR was 0.236 (95% CI 0.125-0.445). However, in stratified analysis by age, the association was 1.058 and 0.662 in participants aged <50 years and aged >50 years respectively. Can you please explain why when looking at participants as a whole the association is stronger than in stratified analysis? Does this mean that aging is the actual risk factor instead of FEV1?

Discussion:

• The discussion should address the discrepancies of results in whole sample and stratified analysis.

• The discussion does not provide implications for future efforts to address hearing loss in population.

6. PLOS authors have the option to publish the peer review history of their article (what does this mean?). If published, this will include your full peer review and any attached files.

Reviewer #1: No

Reviewer #2: No

---

## [Author Response · Author response to Decision Letter 0]

11 Jul 2022

Response to Comments

Thank you very much for providing the comments on critical issues. We worked on the revised manuscript according to the reviewer’s comments. We also worked on English corrections with the help of professional English language editing service. 

Reviewer #1: The submitted manuscript investigates the association between physical function and hearing loss in a Japanese population aged 20-86 years old. The study has great potential, especially due to the underlying data set. However, there are some major revisions required regarding the analyses.

Title

Comment 1: Please indicate the study design in the title.

Response 1: The study design was indicated in the title.

Abstract

Comment 2: The mention of physical activity in the objectives is slightly misleading. The focus should be on physical function.

Response 2: Thank you for mentioning this point. We revised a sentence in the abstract to focus on physical function.

Introduction

Comment 3: Please indicate in ll. 50-51 which population these numbers refer to.

Response 3: Thank you for mentioning it. We added the referring population in the sentence. (ll. 53-54) 

Comment 4: A few more references in the first paragraph would be appropriate.

Response 4: We added three more references in the first paragraph. 

1. Shukla A, Harper M, Pedersen E, Gorman A, Suen JJ, Price C, Applebaum J, Hoyer M, Lin FR, Reed NS. Hearing Loss, Loneliness, and Social Isolation: A Systematic Review. Otolaryngol Head Neck Surg. 2020 May;162(5):622-633. 

2. Deal JA, Reed NS, Kravetz AD, Weinreich H, Yeh C, Lin FR, Altan A. Incident Hearing Loss and Comorbidity: A Longitudinal Administrative Claims Study. JAMA Otolaryngol Head Neck Surg. 2019 Jan 1;145(1):36-43. 

3. Lasak JM, Allen P, McVay T, Lewis D. Hearing loss: diagnosis and management. Prim Care. 2014 Mar;41(1):19-31. 

Comment 5: The introduction focusses primarily on the association of arteriosclerosis and hearing loss. Simultaneously, research results on physical function come short, especially regarding respiratory function. Please elaborate the results of the referenced studies.

Response 5: Thank you for your advice. We further elaborated with references related to physical function, especially for respiratory function in the introduction. (ll. 73-81)

Comment 6: It remains unclear, why it is important to investigate this association and why it should be investigated in a younger population.

Response 6: Thank you for pointing out this omission. We added the explanation with references why it is important to investigate this association and why it should be investigated in younger population. (ll. 83-93) 

Methods

Comment 7: If I understand correctly, you used the average of the maximum values of grip strength in the left and right hand. Instead, using the overall maximum or the maximum of the dominant hand is more appropriate.

Response 7: Thank you very much for pointing this important issue. I found many researchers report the maximum value of dominant hand as you mentioned. We used the dynamometer model of TKK 5401 GRIP D. The mean value of the highest values of the forces of both hands is indicated by flushing after four measurements alternatively measured by each hand on this model. This was a factory setting and measurements followed manufacturer’s instruction. We added about this issue and state as a limitation in discussion. (ll. 401-405) 

Comment 8: I suggest moving the definitions to the respective measurement descriptions.

Response 8: We moved the definitions to respective measurement descriptions. (ll. 146-148 and 150-154)

Comment 9: Please describe how covariates were tested for multicollinearity.

Response 9: This explanation was missing. Thank you for your comment. We added the explanation in the Methods: “The correlation matrix of multivariable parameter estimates was examined to determine if there are serious problems with multicollinearity.” (ll. 182-184)

Comment 10: It is described that hypertension, diabetes, current smoking, and alcohol consumption were added to the models. However, only hypertension is in all three models while diabetes, current smoking, and alcohol consumption were only included for handgrip strength. In addition, variables vary in the age-stratified analyses, and this is not further described. Please specify how your models were derived.

Response 10: The number of subjects were much smaller in VC and FEV1 models. In the first submission manuscript, we reduced the number of independent variables in the models for VC and FEV1. In the revised manuscript, by following the reviewer’s comment on analysis (Comment 20 and 21) and we revised the main analysis Table 3 stratified by sex. To make clear on the discussion for the association between physical function and hearing loss, the same set of variables (age (continuous variable), BMI, current smoking) was used in the models for all types of physical function. Covariate selection was based on the previous information as a risk factor for both predictor (physical function) and outcome (hearing loss). This explanation was added in the methods. Sensitivity analysis by adding other covariates (hypertension, diabetes, alcohol consumption) was also stated in the statistical analysis methods. (ll. 208-217) 

Results

Comment 11: Please indicate how many of the participants were excluded due to not undergoing audiometry or age restrictions and add the response rate.

Response 11: The number of subjects excluded for each reason was stated. (ll. 238-239)

Comment 12: Please add the unit to the standard deviations.

Response 12: The unit to the standard deviations was added. (ll. 244)

Comment 13: The definition of lifestyle variables is only provided in Table 1 and is missing in the methods section.

Response 13: The definitions of lifestyle variables were added in the methods section. (ll. 161-162)

Comment 14: All Tables: Overall n in the top row is missing.

Response 14: Overall n in the top row was added in all tables. 

Comment 15: Table 1: Please also state the number of male and female participants in the respective columns.

Response 15: The numbers of males and females were stated in Table 1.

Comment 16: Table 1: Instead of reporting medication intake, it would be more insightful to report how many were defined as having hypertension, diabetes, or dyslipidemia.

Response 16: We revised Table 1 to report the numbers with defined hypertension, diabetes, or dyslipidemia.

Comment 17: Supplementary material should be referred to only briefly in the results section.

Response 17: We revised to only briefly mentioned for supplementary materials.

Comment 18: Table 2 does not provide highly relevant information and should be moved to the supplements. Instead, it would be interesting to report physical function and hearing loss by age groups. This could also replace S2.

Response 18: We revised Table 2 to report hearing loss and physical function stratified by age group and sex. Initial Table 2 was moved to the supplement and replaced the initial S2. 

Comment 19: Table 3: Due to risk of residual confounding, age should not be included in the analyses as a binary variable but continuous.

Response 19: Thank you for this advice. Age is used as continuous variable in revised Table 3.

Comment 20: Table 3: Participant’s sex appears to be highly relevant. Please consider stratifying your analyses by sex.

Response 20: Thank you for this advice. We did our analysis stratified by sex and presented in the revised Table 3.

Comment 21: Table 4: The age group of over 50-year-olds is still highly heterogeneous regarding physical function and hearing loss. In this context, the age stratification does not seem very relevant. As suggested above, stratifying for sex and including age continuously would deliver more meaningful results.

Response 21: We revised Table 3 with age as continuous variable and stratified by sex. We removed Table 4.

Discussion

Comment 22: In ll. 359-361, it reads as if physical activity is equated with physical function. Please put this into context.

Response 22: Thank you very much for pointing this. This study focused on physical function, not physical activity. We revised this part and discussed only about physical function. (ll. 349-352)

Comment 23: What were the findings of ref. 29? (ll. 361-363)

Response 23: We added the findings of ref. 29 and discussed. (ll. 352-359)

Comment 24: It remains unclear, why associations between physical function parameters and other health outcomes are reported (ll. 370-374).

Response 24: Thank you very much for mentioning this part. It is not relevant. We removed this part in discussion.

Comment 25: How do you explain your unsignificant findings in the younger age group?

Response 25: We removed stratification results by <50 years and ≥50 years, and added sex stratification results as suggested by the reviewer. We found significant findings only in females. We added possible explanation with references. (ll. 332-347) 

Reviewer #2: Thanks for the opportunity to comment on this manuscript. This study is quite interesting which investigated the associations between physical function and hearing loss. Some specific brief comments:

Abstract:

Comment 1: In the method and results of the abstract, the definition of hearing loss should be added.

Response 1: Thank you for pointing it. The definition of hearing loss was added in the method and results of the abstract.

Methods:

Comment 2: Details about how the audiometric test was conducted should be provided. Is bone-conduction used in audiometric testing or not?

Response 2: Detail description of the audiometric test was added. Bone-conduction was not used, and this information is also added. (ll. 114-124)

Comment 3: The definition of hearing loss is >30 dB HL in the methods, whereas in table 1 it says, “Defined as hearing threshold of > 30 dB at 1 kHz and/or > 40 dB at 4 kHz in either ear with pure-tone audiometry”. It is a bit confusing.

Response 3: Thank you very much for pointing this mistake. We stated the same definition in methods and Table 1. (ll. 123)

Comment 4: Can you please explain why BMI and smoking were not adjusted for when looking at the associations between VC and FEV1 and hearing loss?

Response 4: Thank you very much for pointing this issue. The number of subjects were smaller in the models for VC and FEV1 than for handgrip strength. The number of included variables must be reduced for VC and FEV1. Since BMI and smoking have been reported as risks for respiratory function as well as hearing loss, revised Table 3 included BMI and smoking for all types of physical functions. As Reviewer 1 suggested, we revised Table 3 to report the results stratified by sex. 

Results:

Comment 5: When looking at the association between FEV1 and hearing loss in ALL participants, the aOR was 0.236 (95% CI 0.125-0.445). However, in stratified analysis by age, the association was 1.058 and 0.662 in participants aged <50 years and aged >50 years respectively. Can you please explain why when looking at participants as a whole the association is stronger than in stratified analysis? Does this mean that aging is the actual risk factor instead of FEV1?

Response 5: Thank you very much for commenting on this point. This point was also commented by Reviewer 1. The models in initial Table 3 used age category (≥50 years/<50 years) and did not adjust age as continuous variable. This age grouping seems still highly heterogeneous. We revised Table 3 adjusting age as continuous variable. Now we found a significant association only in females. (Revised Table 3)

Discussion:

Comment 6: The discussion should address the discrepancies of results in whole sample and stratified analysis.

Response 6: Now we found insignificant association in males. We added possible explanations with references for the discrepancies of results by sex. (ll. 332-347)

Comment 7: The discussion does not provide implications for future efforts to address hearing loss in population. 

Response 7: Thank you for mentioning it. We provided implications for future efforts to address hearing loss in population. (ll. 411-415)

---

## [Decision Letter · Decision Letter 1]

31 Aug 2022

PONE-D-22-09462R1Hearing loss and physical function in the general population: A cross-sectional studyPLOS ONE

Dear Dr. Wagatsuma,

Thank you for submitting your manuscript to PLOS ONE. After careful consideration, we feel that it has merit but does not fully meet PLOS ONE’s publication criteria as it currently stands. Therefore, we invite you to submit a revised version of the manuscript that addresses the points raised during the review process. Please address all comments raised by the two reviewers.

We look forward to receiving your revised manuscript.

Kind regards,

Karin Bammann, Ph.D.

Academic Editor

PLOS ONE

Journal Requirements:

Reviewers' comments:

Reviewer's Responses to Questions

**Comments to the Author**

1. If the authors have adequately addressed your comments raised in a previous round of review and you feel that this manuscript is now acceptable for publication, you may indicate that here to bypass the “Comments to the Author” section, enter your conflict of interest statement in the “Confidential to Editor” section, and submit your "Accept" recommendation.

Reviewer #1: (No Response)

Reviewer #2: All comments have been addressed

2. Is the manuscript technically sound, and do the data support the conclusions?

Reviewer #1: Yes

Reviewer #2: Yes

3. Has the statistical analysis been performed appropriately and rigorously? 

Reviewer #1: Yes

Reviewer #2: Yes

4. Have the authors made all data underlying the findings in their manuscript fully available?

Reviewer #1: Yes

Reviewer #2: Yes

5. Is the manuscript presented in an intelligible fashion and written in standard English?

Reviewer #1: Yes

Reviewer #2: Yes

6. Review Comments to the Author

Reviewer #1: Thank you for the revised manuscript! I still have a few minor comments:

1. Please report results of male participants in the abstract.

2. ll. 245-252: As suggested, previous Table 2 was moved to the supplementary material. At the same time, the description of this table was not changed and covers an entire paragraph in the results section. Maybe you should move the table back to the results section if you want to describe the results more prominently. Otherwise, I would suggest referencing S2 at an appropriate position without describing the whole table.

3. ll. 264-271: This paragraph explains why all further analyses were stratified by sex. Yet, the analyses before this paragraph were also stratified by sex. I suggest moving the content of this paragraph to the description of statistical analyses in the methods section.

4. A reference for S1 and S4 is missing in the manuscript.

Reviewer #2: Thank you for addressing my comments. Some additional comments:

Introduction

The introduction should focus more on the association between physical function and hearing loss and summarize the gaps of previous studies looking at this association. Finally, you should mention how this study addresses the gaps in previous literature.

Discussion

When comparing the current study with previous studies, it is important to compare the effect sizes of the associations and explain the differences.

7. PLOS authors have the option to publish the peer review history of their article (what does this mean?). If published, this will include your full peer review and any attached files.

Reviewer #1: No

Reviewer #2: No

---

## [Author Response · Author response to Decision Letter 1]

7 Sep 2022

Response to Reviewers

Thank you very much for providing valuable comments to improve the manuscript. We revised according to the reviewer’s comments. 

Reviewer #1: Thank you for the revised manuscript! I still have a few minor comments:

Comment 1: Please report results of male participants in the abstract.

Response 1: The result of male participants was stated in the abstract. (ll. 44)

Comment 2: ll. 245-252: As suggested, previous Table 2 was moved to the supplementary material. At the same time, the description of this table was not changed and covers an entire paragraph in the results section. Maybe you should move the table back to the results section if you want to describe the results more prominently. Otherwise, I would suggest referencing S2 at an appropriate position without describing the whole table.

Response 2: A shortened statement to refer previous S2 Table was re-positioned in ll. 239-242 (new number S4 Table). Supplemental files were re-numbered according to the order of appearance in text.

Comment 3: ll. 264-271: This paragraph explains why all further analyses were stratified by sex. Yet, the analyses before this paragraph were also stratified by sex. I suggest moving the content of this paragraph to the description of statistical analyses in the methods section.

Response 3: Thank you very much for mentioning this point. We agree to shift this part to Methods (ll. 174-178).

Comment 4: A reference for S1 and S4 is missing in the manuscript.

Response 4: A reference for previous S1 (ll. 234; new number S3) and previous S4 (ll. 215; new number S2) was added. 

Reviewer #2: Thank you for addressing my comments. Some additional comments:

Comment 1: Introduction. The introduction should focus more on the association between physical function and hearing loss and summarize the gaps of previous studies looking at this association. Finally, you should mention how this study addresses the gaps in previous literature.

Response 1: Thank you for pointing this crucial point. We agree previous writing of introduction does not flow well into the study objective. As you advised, we revised the introduction to focus more on the association between physical function and hearing loss by revising some sentences and removing lengthy information for the effect of comorbidities such as hypertension and coronary artery disease. We summarized the gaps of previous studies looking at this association and described how this study address the gaps. (ll.72-73; ll.84-92)

Comment 2: Discussion. When comparing the current study with previous studies, it is important to compare the effect sizes of the associations and explain the differences.

Response 2: This is an important comment. We added a paragraph to describe about observed effect size and compare the differences with previous studies in discussion. (ll. 337-348)

---

## [Editor Report · Decision Letter 2]

26 Sep 2022

Hearing loss and physical function in the general population: A cross-sectional study

PONE-D-22-09462R2

Dear Dr. Wagatsuma,

We’re pleased to inform you that your manuscript has been judged scientifically suitable for publication and will be formally accepted for publication once it meets all outstanding technical requirements.

Kind regards,

Karin Bammann, Ph.D.

Academic Editor

PLOS ONE
---

## [Editor Report · Acceptance letter]

28 Sep 2022

PONE-D-22-09462R2 

Hearing loss and physical function in the general population: A cross-sectional study 

Dear Dr. Wagatsuma:

I'm pleased to inform you that your manuscript has been deemed suitable for publication in PLOS ONE. Congratulations! Your manuscript is now with our production department. 

Kind regards, 

on behalf of

PD Dr. Karin Bammann 

Academic Editor

PLOS ONE